# Antifungal Mechanism of Phenazine-1-Carboxylic Acid against *Pestalotiopsis kenyana*

**DOI:** 10.3390/ijms241411274

**Published:** 2023-07-10

**Authors:** Weizhi Xun, Bing Gong, Xingxin Liu, Xiuju Yang, Xia Zhou, Linhong Jin

**Affiliations:** 1Key Laboratory Breeding Base of Green Pesticide and Agricultural Bioengineering, Key Laboratory of Green Pesticide and Agricultural Bioengineering, Ministry of Education, Guizhou University, Guiyang 550025, China; gs.wzxun20@gzu.edu.cn (W.X.); gs.ngong20@gzu.edu.cn (B.G.); gs.xingxinliu21@gzu.edu.cn (X.L.); gs.xjyang20@gzu.edu.cn (X.Y.); xzhou@gzu.edu.cn (X.Z.); 2College of Tea, Guizhou University, Guiyang 550025, China

**Keywords:** bayberry disease, bioactivity, mechanism, *Pestalotiopsis kenyana*, transcriptomics analyses, biogenic fungicide

## Abstract

*Pestalotiopsis* sp. is an important class of plant pathogenic fungi that can infect a variety of crops. We have proved the pathogenicity of *P. kenyana* on bayberry leaves and caused bayberry blight. Phenazine-1-carboxylic acid (PCA) has the characteristics of high efficiency, low toxicity, and environmental friendliness, which can prevent fungal diseases on a variety of crops. In this study, the effect of PCA on the morphological, physiological, and molecular characteristics of *P. kenyana* has been investigated, and the potential antifungal mechanism of PCA against *P. kenyana* was also explored. We applied PCA on *P. kenyana* in vitro and in vivo to determine its inhibitory effect on PCA. It was found that PCA was highly efficient against *P. kenyana,* with EC_50_ around 2.32 μg/mL, and the in vivo effect was 57% at 14 μg/mL. The mechanism of PCA was preliminarily explored by transcriptomics technology. The results showed that after the treatment of PCA, 3613 differential genes were found, focusing on redox processes and various metabolic pathways. In addition, it can also cause mycelial development malformation, damage cell membranes, reduce mitochondrial membrane potential, and increase ROS levels. This result expanded the potential agricultural application of PCA and revealed the possible mechanism against *P. kenyana*.

## 1. Introduction

Pathogenic fungi are known to be one of the lethal diseases in agriculture, which reduce the productivity and quality of plants [1]. In 2004, bayberry blight was first detected and reported, and the disease subsequently spread to various arbutus growing areas, seriously jeopardizing the quality and yield of bayberry [2]. *Pestalotiopsis* is a species-rich asexual genus that produces conidia and is widely distributed in tropical and temperate regions [3]. It is a common plant pathogen and can cause a variety of diseases, including dieback [4], black spot disease [5], grey blight disease [6], and leaf blight [7]. *Pestalotiopsis mangiferae* and *Pestalotiopsis vismiae* have been reported as the main pathogens of bayberry blight [8]. Our previous work found that *Pestalotiopsis kenyana* can also cause the wilting of bayberry leaves.

Occurrence of *P. lushanensis* causing leaf blight on Buddhist pine in China [9]. *P. microspora* can cause leaf spots on moyeam [10]. Leaf spot disease caused by *P. kenyana* on zanthoxylum schinifolium [11]. *P. clavispora* causes brown leaf spots on Chinese bayberry [12]. Shoot blight on Cryptomeria japonica caused by *P. neglecta* [13]. *Pestalotiopsis* sp. causes disease in pinus bungeana [14].

Leaf spot on tea (*Camellia sinensis* (L.) O. Kuntze), caused by *P. trachicarpicola*, can negatively affect the production and quality of tea leaves [15]. Chang Liu et al. [16] revealed the metabolites of *P. kenyana* were significantly different under the induction of *Z. schinifolium* leaves and metabolite 3, 5-dimethoxy benzoic acid, and S-(5-adenosyl)-L-homocysteine were identified as the most likely pathogenic substances. There is no report related to the *P. kenyana* infection in bayberry and their efficient control tragedy and mechanism.

Nowadays, the use of chemical pesticides is still the main means of controlling various diseases. A study suggests that Sodium pheophorbide a (SPA) inhibits spore germination and has detrimental effects on growth, cell wall, membrane permeability, morphology, and enzyme activities of *P. neglecta* mycelia, thereby disturbing the functions of the fungus and strongly inhibiting its growth [17]. Transcriptomic analysis showed most DEGs were involved in the metabolism of amino acids, carbohydrates, and lipids, as well as cell structure and genetic information processing [18]. Pyraclostrobin was reported with significantly high activities in both inhibiting mycelial growth and conidium germination of *P. microspora*, which causes black spot disease on Chinese hickory (Carya cathayensis) [6]. The biogenic ZrONPs showed strong in vitro antifungal activity against *P. versicolor*, by inserting in the *P. versicolor* cell membrane and disrupting the cells of the pathogen [19]. However considerable amounts of harmful pesticide residues often remain in the harvested fruits, becoming a permanent danger to the quality of food, and the environment and can reach the consumer creating health hazards [20].

Phenazine-1-carboxylic acid (PCA) is a naturally secreted product of microbial metabolites of *Pseudomonads* and *Streptomycetes*. The agent can effectively control a variety of bacteria and fungi that harm crops such as rice, melons, fruits, and vegetables and has been registered as a wide-spectrum fungicide in China [21]. PCA can inhibit *Xoo*’s carbohydrate metabolism and energy metabolism, interfere with redox processes in *Xoo*, inhibit catalase (CAT) and superoxide dismutase (SOD) enzyme activities, and lead to increased reactive oxygen species (ROS) accumulation [22]. PCA was found with strong antifungal activity against all phytopathogenic fungi responsible for grapevine trunk diseases and Botryosphaeria dieback and could be proposed as a fungicide against related diseases [23]. As a direct antifungal agent and exopolysaccharide (EPS) inhibitor, PCA has a high potential in the control of *B. cinerea* [24] by reducing NADH and glutathione oxidation, which can occur as cellular oxidation levels increase and mitochondrial instability occurs. As mitochondrial instability decreases, the coupling efficiency on the electron transport chain decreases, resulting in ROS intermediates that lead to cellular oxidative stress and altered membrane permeability. Another study also suggests that PCA can affect the growth and gene expression of *P. infestans* in a time-dependent manner [25]. 

RNA-seq is a sophisticated molecular technique that sheds light on gene expression in cells in physiological and pathological states [26]. High-quality transcriptomes have been sequenced from *P. trachicarpicola*-infected tea leave and acquire a clue to the infection-related gene [15]. Transcriptomic analysis has been employed to unravel potential pathways and genes involved in pecan (*Carya illinoinensis*) resistance to *P. microspora* [17]. The *P. neglecta*’s response to sodium pheophorbide a (SPA) and cell wall degrading enzyme-related gene was revealed in the transcriptomic analysis [25]. 

At present, no data are available on the gene expression profiles of *P. kenyana*, and its response to fungicides. In this study, high-quality transcriptomes and the response were sequenced during interaction using high-throughput sequencing on the Illumina Nova seq 6000 platforms. Because of the lack of reports of biopesticide control of *P. kenyana*, the mechanism of action is not clear. Therefore, it is hoped that PCA, as bio-bactericides, can be applied to bayberry to combat *P. kenyana*. The results of this study show that PCA has limited inhibition of *P. kenyana* in vivo and in vitro. The antifungal mechanism of PCA on *P. kenyana* was studied, and preliminary studies showed that PCA can destroy the structure and function of cell mitochondria, hinder cell energy supply, and thus inhibit the growth of mycelial cells. This result proved the potential of PCA inhibitory effect on the *P. kenyana* in bayberry blight provided the possible antifungal mechanism.

## 2. Results and Discussion

### 2.1. Antifungal Activity

In vitro. Evaluation of the antibacterial activity of PCA against *P. kenyana* was performed in vitro based on hyphal growth rate. It was measured at six concentrations ranging from 0.1 to 20 µg/mL. A solution with no PCA set as control (CK). Compounds showing a clear dose-response with the regression equation Y = 0.1573x + 0.1347, R^2^ = 0.9699, then had EC_50_ values determined as 2.32 µg/mL (Figure 1a,b). This result indicated that PCA could significantly inhibit the growth of hyphae of *P. kenyana* in vitro.

In vivo. The antifungal activity data for PCA in vivo is shown in Figure 1c. PCA had good control effects on *P. kenyana*, and the inhibitory effects on *P. kenyana* at test concentrations were 57 ± 5.3%. 

### 2.2. Effects on the Morphology of P. kenyana

For fungi, the normal morphology of the mycelium is a key factor affecting the vitality fungus [27]. As shown in Figure 2, the control group (Figure 2a,c) had smooth hyphae morphology and normal development. After *P. kenyana* mycelium was treated with PCA (3 μg/mL), the surface of the hyphae was rough, wrinkled, and deformed, and some of them swelled and deformed in Figure 2b,d. Then, TEM was performed and compared with normal cells in the control group (Figure 2e), the boundaries of each organelle were blurred, the mitochondrial matrix was swollen, and some lipid droplets appeared after being treated by PCA (Figure 2f). The mitochondrial matrix was swollen and extended. The organelles in the cytoplasm were disorganized, the endoplasmic reticulum was expanded. Additionally, lipid droplets (Ld) congregated to raise size and number.

### 2.3. Effect on ROS Production and Mitochondrial Membrane Potential

Reactive oxygen species (ROS) are a group of unstable and highly reactive molecules or free radicals [28], molecular oxygen is produced through cellular metabolism, derived from mitochondria. The results showed that compared with the control group (Figure 3a), the fluorescence signal strengthens significantly after PCA treatment indicating the amount of ROS in *P. kenyana* mycelial cell was significantly increased (Figure 3b). While at high ROS levels, the balance between ROS production and the antioxidant systems is lost, resulting in oxidative stress. This subsequently destroys intracellular biomacromolecules (such as nucleic acids, membrane lipids, and cellular proteins) and Oxidative damage to these biomolecules triggers apoptosis [29]. These results suggest that PCA may induce the generation and accumulation of ROS which subsequently affect mitochondrial structure and function. The unique lipid composition and structure of the mitochondrial membrane are essential for the proper function of mitochondria [30]. Mitochondria are the main producers of ROS, and the accumulation of intracellular ROS can have serious effects on cells [31]. A low concentration of ROS could affect signal transduction, but a high concentration could lead to the dysfunction of MMP and cell death. The above results showed that PCA may affect the mitochondrial structure and functions. To demonstrate this mode of action again, we investigated the effect of this compound on the MMP of *P. kenyana* mycelial. Figure 3c,d showed that the fluorescence intensity of the dye was reduced compared with the control group where the MMP was significantly decreased. These results suggest that PCA can damage the MMP and lead to mitochondrial dysfunction in *P. kenyana* mycelia. Mitochondria play an important role in energy metabolism and apoptosis, and changes in the mitochondrial structure and functions of mycelium would affect cell respiratory activity [32].

### 2.4. Effect on the Cell Membrane Permeability and Cellular Leakage

The integrity of the cell membrane plays a crucial role in the growth and development of the fungus. Some studies have shown that ozone, as a strong oxidant acts on the cell membrane of pathogenic fungi, destroys the integrity of the cell membrane, causes the outflow of intracellular substances, and thus inhibits the growth of pathogenic bacteria [33]. The cell membranes of pathogenic microorganisms are the first barrier against many microbicides and are the target of various evolutionary strategies to promote survival and infection [34]. In this study, remarkable morphological changes of *P. kenyana* were found in PCA treatment. Therefore, this study also demonstrates the destruction of *P. kenyana* cell membrane integrity by PCA from the perspective of relative conductivity. As shown in Figure 4a, the relative conductivity of *P. kenyana* cell membranes increases with the increase of treatment time and the concentration of the agent, showing time-dependent and concentration-dependent. There was no significant discrepancy in the relative conductivity between the control group and 1 μg/mL of PCA after 6 h. When the PCA concentration was 5 and 10 μg/mL of PCA, the discrepancy of relative electric conductivity was not significant in primal 2 h, but with time going by, the relative electric conductivity was significantly higher than all the lower concentrations. It showed that PCA had certain damage to the cell membrane.

In addition to relative conductivity, nucleic acids, and proteins can be used as membrane permeability parameters to determine whether the cell membrane is damaged, with UV absorption maxima at 260 nm (OD_260_) and 280 nm (OD_280_), respectively [35]. As shown in Figure 4b, 4c, OD_260_ and OD_280_ were both increased in a time-dependent and concentration-dependent manner. In the primal 2 h, there was no significant difference in the absorption values of the four treatment groups. After 8 h of treatment, OD_260_ and OD_280_ were significantly higher than those in the blank control group (*p* < 0.05), but no noticeable difference between 1 and 5 μg/mL of PCA. These results further demonstrated that the cell membrane integrity of *P. kenyana* had been damaged by PCA.

### 2.5. Effect on Nuclear Morphology of P. kenyana

Subsequently, the effect of PCA on the nuclear morphology of *P. kenyana* was evaluated. The fluorescence experiments showed that PCA can also interact with the nucleus of the hyphal cell and diminish the fluorescence intensity. In addition, the size of the nucleus of the hyphal cell treated with PCA became smaller and the nuclear number decrease significantly compared to the control group (Figure 5). As reported, the geometry of the nucleus has a significant effect on cell proliferation, gene expression, protein synthesis, and normal physiological aging and pathological conditions which in turn can change the size and shape of the nucleus [36].

From the above figures, it can be inferred that PCA might affect the cell proliferation of *P. kenyana* mycelia by function destruction of various cellular components and hence inhibiting cell proliferation.

### 2.6. Transcriptome Analysis

Transcriptome sequencing of two treated hyphae samples yielded data of 13.7 GB, respectively. Transcriptome in the PCA-treated hyphae was found 2236-genes-upregulated and 1377-genes-downregulated when compared to the control group. The differential gene expression (DGE) between PCA-treated *P. kenyana* cells and normal cells (CK) was illustrated in Figure 6.

The results of Gene Ontology (GO) enrichment analysis between the PCA-treated group and the control group (Figure 7a) showed that for the biological process (BP), the differential genes were significantly enriched in the oxidation-reduction process, metabolic process, and transmembrane transport. In terms of cellular component (CC), the nucleus, membrane components, cytoplasm, and enrichment were significant; In terms of molecular function (MF), the differential genes were mainly enriched in oxidoreductase activity, catalytic activity, zinc ion binding, and heme binding. 

Kyoto Encyclopedia of Genes and Genomes (KEGG) enrichment analysis showed that after PCA treatment (Figure 7b), differential genes were mainly enriched in amino acid and carbohydrate metabolic pathways. For example, glycine, serine, and threonine metabolism (53), starch and sucrose metabolism (51), tyrosine metabolism (47), tryptophan metabolism (44), and phenylalanine metabolism (38). 

Combining GO enrichment analysis and KEGG enrichment analysis, it was found that most of the significantly different genes were related to mitochondrial structure and function. Mitochondria, as highly dynamic major organelles in eukaryotes, are responsible for the body’s energy metabolism [37]. Changes in mitochondrial structure and function affect the normal functioning of their functions, resulting in reduced energy production and hindered cellular activity. Pyruvate is an organic acid that plays a key role in central metabolic pathways [38]. In eukaryotes, pyruvate produced by glycolysis is used for conversion to ethanol and lactic acid and anabolism in the cytoplasm or is transported into mitochondria for use as a substrate in the tricarboxylic acid (TCA) cycle [39]. We investigated the expression of genes that differ in pyruvate metabolism (Figure 8).

### 2.7. Gene Expression Analysis

qRT-PCR assay was performed on four genes related to pyruvate metabolism, including *erg10*, *cut6*, *lys1*, and *acu-6*, and the result is shown in Figure 9. Among them, only the expression of the *acu-6* gene was up-regulated, and the expression of the remaining three genes was down-regulated. This is consistent with transcriptome sequencing results and confirms the reliability of transcriptomics data. This result indicated PCA had an impact on pyruvate metabolism and caused a decline in energy metabolism. 

## 3. Materials and Methods

### 3.1. Biomaterials

All the commercial bactericides used in our experiments are commercially available. Phenazine-1-carboxylic acid (purity 98%) was sourced from Zhengzhou Alpha Chemical Co., Ltd. (Zhengzhou, China). The *P. kenyana* strain tested (ACCC 35163) was provided by our group and stored in 30% glycerol at −80 °C in a freezer. We used the conventional media for *P. kenyana* culture potato dextrose agar (PDA) medium and potato dextrose broth (PDB) medium and both were purchased from Beijing Solarbio Technology Co., Ltd. (Beijing, China).

### 3.2. Antifungal Assay

In vitro. The inhibitory effect of PCA on *P. kenyana* hyphae growth rate was tested with PCA in a concentration gradient 0.1, 1, 2.5, 5, 10, 20, 50, 100 μg/mL dissolved in dimethyl sulfoxide 0.5% DMSO (*v*/*v*). Solution with no compound PCA was taken as control. The test strain was *P. kenyana* (ACCC 35163), which was isolated and identified by our group from withered bayberry leaf samples in Wencheng County, Zhejiang China. Mycelial plugs (5 mm diameter) from the leading edge of a 5-day-old colony) were placed on a series of plates with new PDA containing different concentrations of PCA. After incubation for 5 days at 25 °C in darkness, the diameter of each colony was measured. The inhibitory effect of PCA against *P. kenyana* is expressed by the following formula [40].
$$\mathrm{Inhibition}\;\mathrm{rate}=\frac{\left(C-5\right)-(S-5)}{C-5}\times 100\%$$

In the formula, C and S are the average diameters (mm) of colonies in the control group and the treatment group respectively, and 5 mm was the diameter of the mycelia plug.

In vivo. The presence or absence of PCA on *P. kenyana* was determined using a syringe penetration method. Two-year-old Dongkui bayberry was selected as the experimental material and kept under an artificial climate chamber (25 ± 0.4 °C, humidity 80%) for three months. The conidial suspensions (10^6^ conidia mL^−1^) of *P. kenyana* were injected using a sterilized syringe at the center of the leaf. Two days after inoculation, 1% PCA suspension concentrate was sprayed on the infected leaf with active ingredient 14 μg/mL (700 times dilution of SC), and the leaves were sprayed with sterile water as a control. After 10 days, the inhibition effect was calculated by the following formula [28].
$$\mathrm{Antifungal}\;\mathrm{activity}=\frac{C-T}{C}\times 100\%$$
where C is the disease diameter of the control group and T is the disease diameter of the treatment group.

### 3.3. Histocytological Assay

The morphological change of *P. kenyana* treated by PCA was detected by stereomicroscope, scanning electron microscope (SEM), and transmission electron microscopy (TEM).

Mycelial plugs (5 mm diameter) from the leading edge of a 5-day-old colony of *P. kenyana* on PDA were transferred to PDB and cultured on a shaking incubator (Shanghai Tiancheng Experimental Instrument Manufacturing Co., Ltd., Shanghai, China) at 25 °C, 180 rpm for 3 days, at which PCA was added to a final concentration of 3 µg/mL. Mycelial without PCA was used as the blank control. 24 h later, hyphal morphology was observed under the stereomicroscope (VHX-6000, Keyence, Osaka, Japan).

For SEM observation, the appropriate changes were made according to the method of Chen [41]. After treating the hyphae with PCA (3 μg/mL) for 24 h, fix the mycelium overnight with 2.5% glutaraldehyde at 4 °C overnight and washed three times with PBS (phosphate-buffered saline at pH 7.4) for 5 min each. Afterward, samples were dehydrated by increasing graded ethanol (50%, 70%, 80%, 90%, 95%, and 100%) for 15 min at each step then were transferred to an ethanol-isoamyl acetate mixture (*v*/*v* = 1:1) for 15 min and isoamyl acetate at least 30 min, and then critical-point dried (K850 Critical-Point Dryer; Quorum Technologies, Lewes, United Kingdom) for observing the hyphae morphology under SEM (Hitachi U8010, Tokyo, Japan).

The TEM assay was performed by following steps. PCA-treated mycelium plugs were rinsed with PBS and then dehydrated through a series of ethanol (50, 70, 80, 90, 95, 100, and 100%). The sample was then fixed in acetone and 812 embedding mixture (1:1) and 100% 812 embedding agents overnight. After polymerization at 60 °C for 48 h, the sample was subjected to a microtome to produce 60–80 nm ultra-thin slices with a diamond knife (UC7; Leica, Wetzlar, Germany). After drying at a critical point using uranyl acetate, finally, the sample was observed and photographed by TEM (H-7650; Hitachi, Tokyo, Japan) [42].

### 3.4. Reactive Oxygen Species (ROS) Accumulation and Mitochondrial Membrane Potential Assay

After 24 h of treatment of PCA (3 μg/mL), *P. kenyana* mycelia were picked and placed on a sterile slide. Then, 2′, 7′-dichloro-dihydro-fluorescein diacetates (DCFH-DA) and rhodamine 123 solutions (purchased from Beyotime, Shanghai, China) were used to stain mycelium for ROS and MMP, respectively. The samples were incubated at 37 °C for 30 min in darkness and then washed with PBS (pH 7.4) three times, then the samples were photographed using fluorescence microscopy (FVMPE-RS; Olympus, Tokyo, Japan). The fluorescence intensity of the stained mycelia was recorded for evaluating and comparing the ROS production and MMP of the different groups [43]. 

### 3.5. Observation of the Cell Membrane of P. kenyana

Measurement of Membrane Permeability. The cell membrane relative permeability rate of *P. kenyana* was evaluated according to the described method [44]. Mycelial plugs (5 mm diameter) taken from the 3-day-old colony were cultured at 25 °C with shaking at 180 rpm in a PDB medium for 3 days. Then the mycelia were harvested and washed thrice with PBS buffer (pH 7.2–7.4). Afterward, 100.00 mg of mycelia were suspended in 10 mL of sterile distilled water treated with various concentrations of PCA. Finally, the conductivities of the solution were determined with a conductivity detector (HORIBA EC1100, Kyoto, Japan) (at 0 h was marked as L0 and at 0.5 h~24 h was marked as L1). The final electric conductivities were measured after the mycelia were boiled for 30 min and remarked as L2. The relative permeability rate of mycelia was calculated by the following formula:$$Relative\;electric\;conductivity=\frac{L1-L0}{L2-L0}\times 100\%$$

Determination of Cellular Leakage. The effects of PCA on cellular leakage were performed according to the previously described methods with some modifications [45]. *P. kenyana* was cultured in a PDB medium to obtain mycelia. Then, PCA solution (1, 5, and 10 μg/mL) mixed with PBS was added to the centrifuge with mycelia to obtain hyphae suspension. Then, the samples were incubated at 30 ± 1 °C for 2 h, 4 h, 6 h, and 8 h, and then centrifuged. Finally, the absorbance values of the supernatants were measured at 260 nm and 280 nm using a microplate reader (Bio-Tek Synergy2, Winooski, VT, USA), with three repeats.

### 3.6. Nuclear Staining Assay

The effect on the nuclei of *P. kenyana* mycelia was performed according to the method previously reported [46]. The hyphae of *P. kenyana* treated with PCA (3 μg/mL) were fixed with stain fixative for 30 min at 4 °C and washed twice with 0.01 M PBS, then stained with 0.5 mL of Hoechst 33258 solution (purchased from Beyotime, China) at 25 °C for 20 min. After incubation, the samples were rinsed twice with PBS. At last, a drop of fluorescence quencher was added and mounted on a cover glass to be photographed under a fluorescence microscope, immediately (FVMPE-RS; Olympus, Tokyo, Japan).

### 3.7. Transcriptome Analysis

The 5 mm mycelia were placed in PDB medium and shaken at 180 rpm for 24 h at 25 °C in an incubator (Shanghai Tiancheng Experimental Instrument Manufacturing Co., Ltd.) for 3 d, and then PCA was added to the broth at a final concentration of 3 μg/mL. After 24 h treatment, the mycelia were harvested and stored at −80 °C. The mycelia grown in a PCA-free solution was taken as a control.

Total RNA was extracted using OMEGA Fungal RNA Kit following the manufacturer’s procedure. The total RNA quantity and purity were analyzed of Bioanalyzer 2100 and RNA 1000 Nano LabChip Kit (Agilent, Santa Clara, CA, USA) with RIN number >7.0. Poly(A) RNA is purified from total RNA (5 ug) using poly-T oligo-attached magnetic beads using two rounds of purification. Following purification, the mRNA is fragmented into small pieces using divalent cations under elevated temperatures. Then the cleaved RNA fragments were reverse-transcribed to create the final cDNA library by the protocol for the mRNASeq sample preparation kit (Illumina, San Diego, CA, USA), the average insert size for the paired-end libraries was 300 bp (±50 bp). We then performed the paired-end sequencing on an Illumina Novaseq™ 6000 at the (LC Sciences, San Diego, CA, USA) following the vendor’s recommended protocol.

Firstly, Cutadapt [47] and Perl scripts in-house were used to remove the reads that contained adaptor contamination, low-quality bases, and undetermined bases. The sequence quality was verified using FastQC (http://www.bioinformatics.babraham.ac.uk/projects/fastqc/) (accessed on 15 December 2022). including the Q20, Q30, and GC content of the clean data. All downstream analyses were based on clean data of high quality. De novo assembly of the transcriptome was performed with Trinity 2.4.0 [48]. Trinity groups transcripts into clusters based on shared sequence content. Such a transcript cluster is very loosely referred to as a ‘gene’. The longest transcript in the cluster was chosen as the ‘gene’ sequence (aka Unigene).

All assembled Unigenes were aligned against the non-redundant (Nr) protein database (http://www.ncbi.nlm.nih.gov/) (accessed on 16 December 2022), Gene ontology (GO) (http://www.geneontology.org) (accessed on 16 December 2022), SwissProt (http://www.expasy.ch/sprot/) (accessed on 17 December 2022), Kyoto Encyclopedia of Genes and Genomes (KEGG) (http://www.genome.jp/kegg/) (accessed on 17 December 2022) and eggnog (http://eggnogdb.embl.de/) (accessed on 18 December 2022) databases using DIAMOND [49] with a threshold of E value <0.00001.

Salmon [50] was used to perform expression levels for Unigenes by calculating TPM [51]. The differentially expressed Unigenes were selected with log2 (fold change) >1 or log2 (fold change) <−1 and with statistical significance (*p*-value < 0.05) by R package edgeR [52].

### 3.8. qRT-PCR Analysis

After washing the mycelia with phosphate-buffered saline (PBS) twice, the total RNA of *P. kenyana* mycelia treated with PCA (3 μg/mL) was extracted by the OMEGA Fungal RNA Kit (Omega Bio-Tek, Guangzhou, China). Then, RNA was reverse transcribed into cDNAs according to the M-MLV First Strand cDNA Synthesis Kit (Omega Bio-Tek, Guangzhou, China). Finally, the perfect start SYBR Green qPCR master mix was used for amplification. The reaction conditions are 95 °C for 10 min hold, 95 °C for 5 s, 60 °C for 20 s, 72 °C for 20 s, and amplification for 40 cycles. The primer sequences used for the analysis are shown in Table 1. Actin was used as an internal control for normalization. The relative expression level of each gene was calculated by the 2^−ΔΔCT^ method. All tests were performed in triplicate.

### 3.9. Statistical Analysis

All statistical analyses were performed using IBM SPSS Statistics 26. Bioassay data were analyzed with Log_10_ probity analysis to obtain EC_50_ values. All data were presented as mean ± standard deviation and assessed by one-way ANOVA. Statistical significance was considered at *p* < 0.05.

## 4. Conclusions

In summary, we investigated the control effect and mechanism of PCA against *P. kenyana*. in bayberry. We found that PCA has good antifungal activity against *P. kenyana*. PCA can destroy the cell membrane and inhibit the formation of the nucleus. In addition, it causes the deformation of hyphae structure and mitochondrial functioning, increment of ROS production, destruction of MMP, and all features together overwhelmingly inclined toward inhibiting the growth of *P. kenyana*. Further, the transcripts analysis and function prediction revealed that PCA significantly down-regulated pyruvate metabolism genes and impaired energy metabolism. These results expanded the potential agricultural application of PCA and revealed the possible action mechanism against *P. kenyana*.

## Figures and Tables

**Figure 1 ijms-24-11274-f001:**
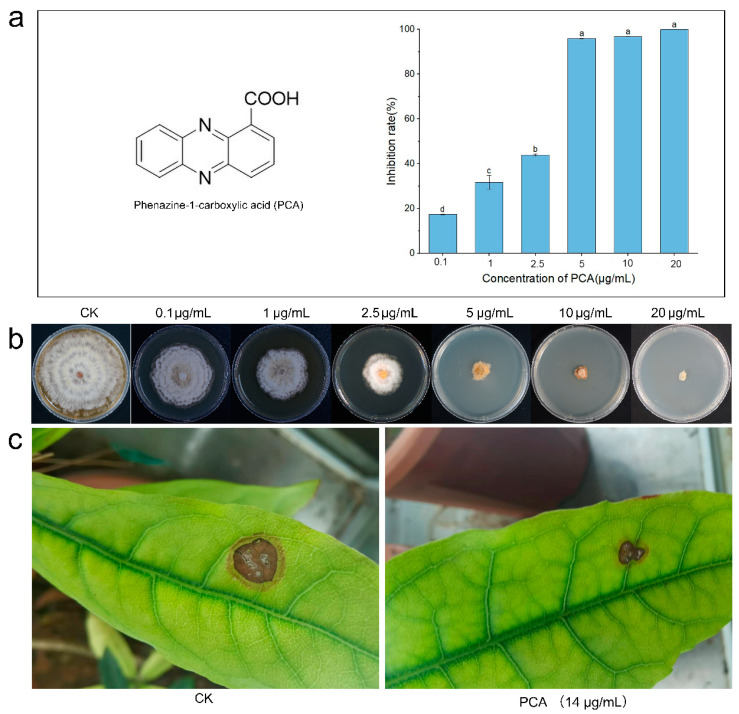
Antifungal activity of PCA against *P. kenyana* in vivo and in vitro. (**a**) The structure of PCA and the inhibition rates of PCA on *P. kenyana* hyphae growth. Data are displayed as the mean ± SD. (**b**) In vitro test of PCA against *P. kenyana* at different concentrations. (**c**) In vivo antifungal activity of PCA (700 times dilution of 1% PCA suspension concentrate, 14 μg/mL) against *P. kenyana*. CK means control group treated with water (no PCA). Value represents the mean ± SD (*n* = 3), different letters indicate significant differences (*p* < 0.05).

**Figure 2 ijms-24-11274-f002:**
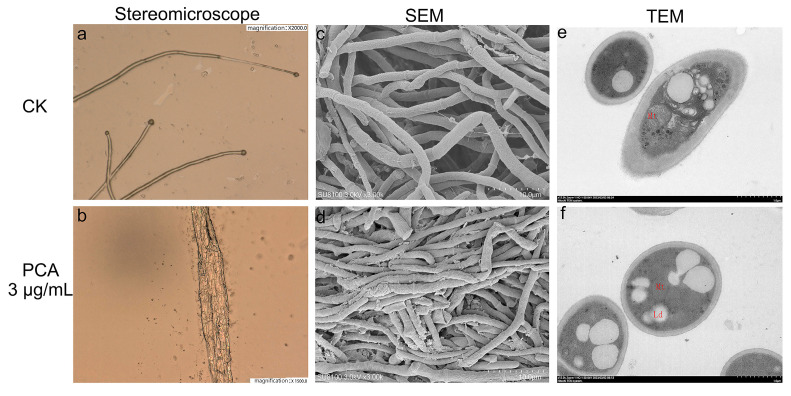
Morphological detection of *P. kenyana* hyphae observed by stereomicroscope, scanning electron microscope (SEM), and transmission electron microscope (TEM) (images (**a**,**c**,**e**) in up line is the sample of the control group, and the lower ones (**b**,**d**,**f**) are samples treated by PCA at 3 μg/mL. The red letter “Mt” and “Ld” in e and f represent Mitochondria and Lipid droplets, respectively.

**Figure 3 ijms-24-11274-f003:**
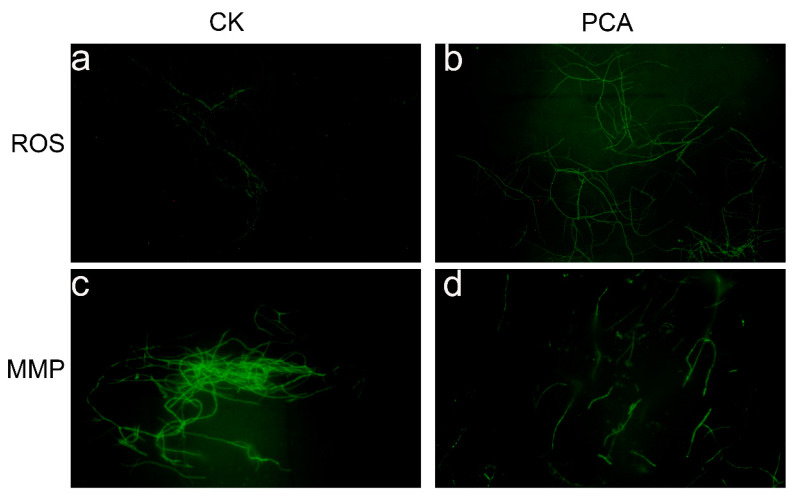
Fluorescence micrographs of *P. kenyana* mycelial, ROS stained with DCFH-DA (**a**), control group with no PCA(CK); (**b**), PCA-treated group; MMP stained with Rhodamine 123, (**c**): control group with no PCA(CK); (**d**): PCA-treated group.

**Figure 4 ijms-24-11274-f004:**
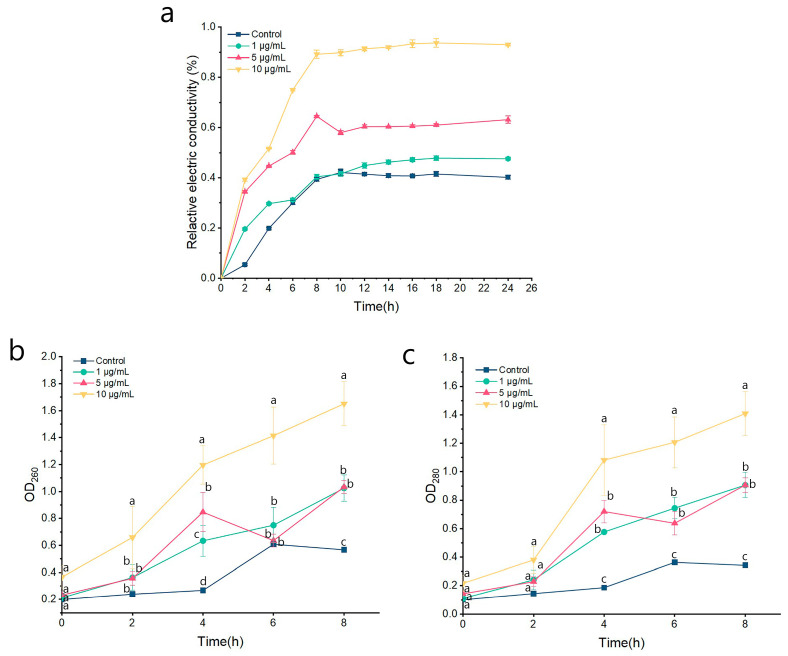
Effects of PCA on cell membrane permeability and cell leakage of *P. kenyana*. (**a**) Cell membrane permeability. (**b**) The absorbance value of nucleic acids (OD_260_). (**c**) The absorbance value of protein (OD_280_). Data are displayed as the mean ± SD of three replications. Lines with the “a, b, c” indicate significant differences (*p* < 0.05) for different concentrations.

**Figure 5 ijms-24-11274-f005:**
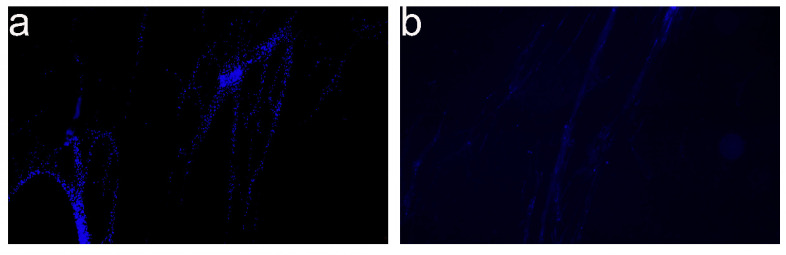
Fluorescence micrographs of *P. kenyana* nuclear stained with Hoechst 33258 to evaluate the effects of PCA on nuclear at 3 μg/mL. (**a**): control group with no PCA(CK); (**b**), PCA-treated group.

**Figure 6 ijms-24-11274-f006:**
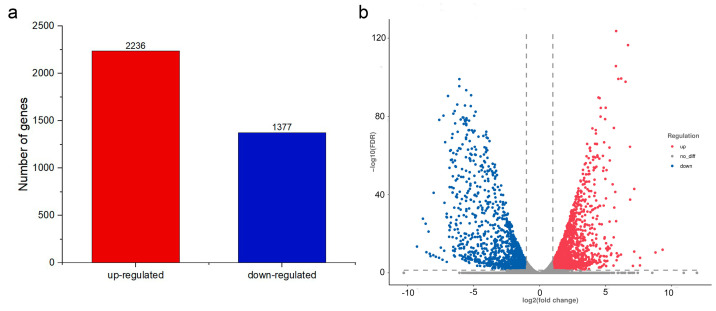
DGE identified by transcriptomics. (**a**) The statistical plot of the number of differential genes between the PCA group and the control group (CK). (**b**) Volcano plot of DGE. The absolute value of fold change (FC) was greater than 2-fold, and the corrected *p*-value was q-value <0.05. Red: upregulated genes, blue: downregulated genes.

**Figure 7 ijms-24-11274-f007:**
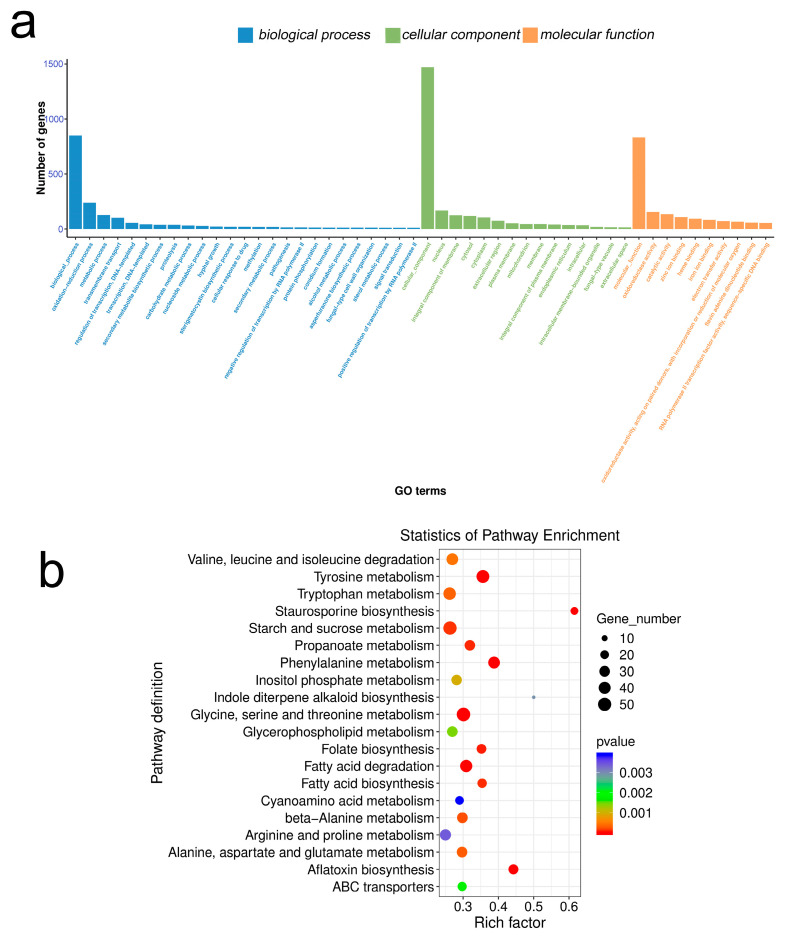
GO enrichment classification histogram and KEGG enrichment bubble chart. (**a**) GO enrichment classification histogram; different colors represent different categories. (**b**) KEGG enrichment bubble plot; the rich factor refers to the ratio of the number of DEGs in the pathway and the number of all annotated genes in the pathway. The red arrows refer to the functional and metabolic pathways associated with mitochondria.

**Figure 8 ijms-24-11274-f008:**
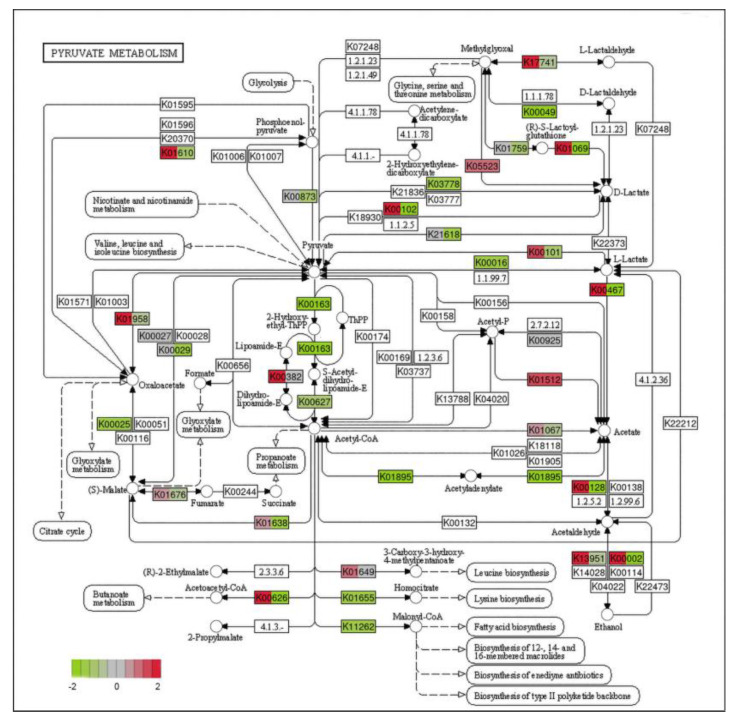
Diagram of pyruvate metabolism pathway in PCA-treated group (green represents down-regulated, and red represents up-regulate).

**Figure 9 ijms-24-11274-f009:**
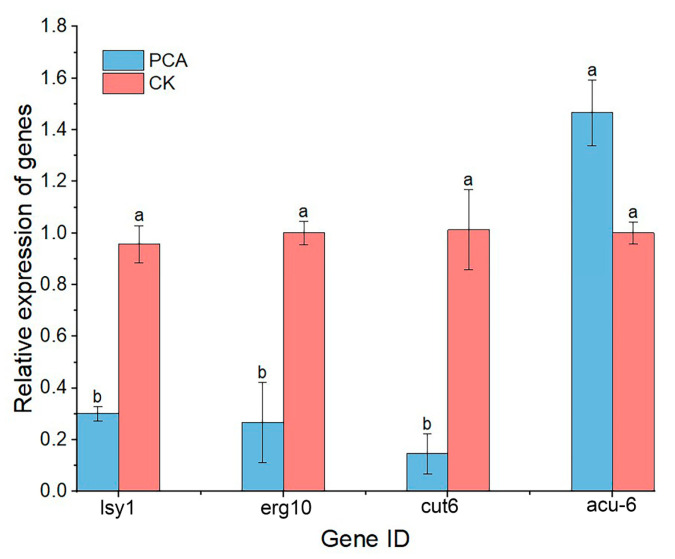
Relative expression of genes that differ in pyruvate metabolism. Data are displayed as the mean ± SD. Data are displayed as the mean ± SD of three replications. Different letters indicate significant differences. CK means the data from the control group that normal *P. kenyana*. (with no PCA treatment).

**Table 1 ijms-24-11274-t001:** Differentially Expressed Genes and Their Primer Sequences in qRT-PCR.

Gene_ID	log_2_FC	Primer Sequences (5′–3′)	Primer’s Efficiency
acu-6	2.03	F-TGTATACGAAGTCGGTTCAGCC R-TAACCGTCAACGACGTAAATTC	96
erg10	−1.73	F-TCCAACACCCCGCACTACCTGC R-GCTGACCTCGACGGGGACGATC	95
lys1	−1.31	F-GTGAACAGTTCGCCAATGCCTA R-AGAGTGCTCGCGGAGGTGCTG	97
cut6	−1.46	F-AGCAGCCTCCCCAGGGTGAAT R-TCGGTACACTTGTTGAAGAAGTGG	97
actin		F-CTGGCACCGTCGTCGATGAAG R-AAGGTCCGCTCTCGTCGTACTC	99

## Data Availability

Not applicable.

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
