# Peer review of "Antifungal Mechanism of Phenazine-1-Carboxylic Acid against Pestalotiopsis kenyana"

_ijms, 2023, doi:10.3390/ijms241411274_

Round 1

Reviewer 1 Report

This research is under the scope of this journal; the topic is relevant for readers, and this research deals with potentially significant knowledge to the field. And It will be important of Microbiology knowledge. The topic is relevant for readers and this study deals with potentially significant knowledge to the field and open new way for future studies. 

Please add more keywords, and order the keywords / Mesh terms alphabetically

Please consider explain better limitations.

Please consider other references from 2023

Sample size calculation is not clear. Please better describe the primary outcome utilized, standard deviation and the mean average among groups.

Check references

Please, add more information in figure legends

Author Response

Responses to the Reviewer 1

Dear reviewer, we are so appreciated your earnest review and great advice. We have taken care of all the revisions accordingly.

Reviewers 1

  1. Please add more keywords, and order the keywords / Mesh terms alphabetically

Responses: Thanks for your valuable comments and suggestions. This section has been modified in the manuscript according to your comments. "Keywords: Bayberry disease, Bioactivity , Mechanism, Pestalotiopsis kenyana, Transcriptomics Analyses, Biogenic fungicide.

  1. Please consider explain better limitations.

Responses: Thanks for your valuable comments and suggestions. What we want to express is that PCA has good inhibitory activity in vitro and slightly weaker inhibitory activity in vitro.

  1. Please consider other references from 2023

Responses: Thanks for your valuable comments and suggestions. It has been modified in the manuscript according to your comments. We added some new references to replace the older ones in the manuscript.

  1. Sample size calculation is not clear. Please better describe the primary outcome utilized, standard deviation and the mean average among groups.

Responses: Thanks for your valuable comments and suggestions. It has been modified in the manuscript according to your comments.  "Data are displayed as the mean ± SD of three replications." was added followed the figures1, 4

  1. Check references

Responses: Thanks for your valuable comments and suggestions. We have changed some references in the manuscript.

  1. Please, add more information in figure legends

Responses: Thanks for your valuable comments and suggestions. It has been added more information in the manuscript according to your comments.

Reviewer 2 Report

The manuscript submitted to International Journal of Molecular Sciences investigates the influence of PCA on the physiology, morphology and transcriptome of P. kenyana, which is one of the main fungi affecting bayberry. In addition, the authors attempted to assess the effect of PCA on the regulation of the expression of selected genes involved in pyruvate metabolism.

In my opinion the abstract and introduction to the manuscript outline the research topic in its entirety, but requires some editorial and linguistic correction. The results are described in detail, but the discussion is conducted quite briefly, I suggest enriching it with more citations concerning other phenolic substances with antifungal activity. The work is interesting, but it contains minor deficiencies (especially editorial and language), that should be corrected in the manuscript before it is published. Moreover, the manuscript contains inconsistencies between the individual parts, e.g. EC50 value in the abstract and the description of the results.

Below are my specific comments:

1.     Unification of the spelling of Latin names with cursive throughout the work. The same applies to the use of the words in vitro, in vivo, de novo.

2.     Linguistic and editorial rewording of the introduction - in particular, the paragraph in lines 35-42.

3.     Why the EC50 value determined by the authors differs in the abstract (2.32 µg/mL) from the one given in the description of the results (0.87 µg/mL PCA). Please correct it.

4.     Section 2.2 should start on a new page.

5.     Description of Figure 2 - shouldn't the word microscope be used instead of micrograph when describing TEM?

6.     In the case of figures 4-9, although they are colored, they are so small that their correct reading and interpretation of the data are difficult. I suggest enlarging them.

7.     Why the Authors used different concentrations of PCA in different experiments? This makes interpretation of the results difficult. Why Authors chose a PCA concentration of 3 µg/mL for most experiments and 1, 5 and 10 µg/mL for others. Please explain.

8.     Descriptions of figures - if there is information about statistical data in the figures, marked e.g. as a, b, etc., then it should be explained in the description of the figures. Please complete it.

9.     Subsection 3.1 - I suggest changing the name to “Chemicals and strains”, because biomaterials refers to a completely different group of materials used, among others, for the production of implants, tissue scaffolds, etc.

10.  Please read the entire manuscript carefully to remove various editorial errors, e.g. Line 404 - repetition of the word The

Considering all the above-mentioned aspects, this manuscript cannot be published in International Journal of Molecular Sciences in present form and requires major improvements.

Author Response

Responses to the Reviewer 2

Dear reviewer, we are so appreciated your earnest review and great advice. We have taken care of all the revisions accordingly.

  1. Unification of the spelling of Latin names with cursive throughout the work. The same applies to the use of the words in vitro, in vivo, de novo.

Thanks for your valuable comments and suggestions. We have modified the spelling according to your comments.

  1. Linguistic and editorial rewording of the introduction - in particular, the paragraph in lines 35-42.

Thanks for your valuable comments and suggestions. We have changed the order of presentation of lines 5-42 according to your comments.

  1. Why the EC50value determined by the authors differs in the abstract (2.32 µg/mL) from the one given in the description of the results (0.87 µg/mL PCA). Please correct it.

Thanks for your valuable comments and suggestions. That was a typo and has been corrected in the manuscript.

  1. Section 2.2 should start on a new page.

Thanks for your valuable comments and suggestions. We have restarted section 2.2 on a new page with your comments.

  1. Description of Figure 2 - shouldn't the word microscope be used instead of micrograph when describing TEM?

Thanks for your valuable comments and suggestions. We have corrected the incorrect formulation in accordance with your comments.

  1. In the case of figures 4-9, although they are colored, they are so small that their correct reading and interpretation of the data are difficult. I suggest enlarging them.

Thanks for your valuable comments and suggestions. We have enlarged the picture as you suggested

  1. Why the Authors used different concentrations of PCA in different experiments? This makes interpretation of the results difficult. Why Authors chose a PCA concentration of 3 µg/mL for most experiments and 1, 5 and 10 µg/mL for others. Please explain.

Thanks for your valuable comments and suggestions. Most of the experimental designs were performed at EC50 (2.32 µg/mL) concentrations, and the reason for using 3 is that the reagent concentration configuration is more accurate. The experiment selected 1.5.10 µg/mL was chosen to demonstrate the time-dependent dependence of the PCA's action according to the method in the literature.

  1. Descriptions of figures - if there is information about statistical data in the figures, marked e.g. as a, b, etc., then it should be explained in the description of the figures. Please complete it.

Thanks for your valuable comments and suggestions. We've added relevant information to the captions.

  1. Subsection 3.1 - I suggest changing the name to “Chemicals and strains”, because biomaterials refers to a completely different group of materials used, among others, for the production of implants, tissue scaffolds, etc.

Thanks for your valuable comments and suggestions. We completely agree with you and have changed the name of Section 3.1 to “Chemicals and strains”.

  1. Please read the entire manuscript carefully to remove various editorial errors, e.g. Line 404 - repetition of the word The

Thanks for your valuable comments and suggestions. We have removed the extra words.

Reviewer 3 Report

The paper submitted to the IJMS entitled " Integrated Transcriptomics Analyses, Physiology and Morphology Reveals the Possible Antifungal Mechanism of Phenazine-1-carboxylic Acid Against Pestalotiopsis kenyana." By Weizhi Xun et al., present original research results that are important for practice because it concerns the formation of biogenic fungicides.

Based on my thorough evaluation of the document, I am certain that it holds significant value and can be authorized for publishing upon completion of the required revisions.

Comments:

1. I think you should consider changing the title of the paper because it is long and contains redundant information. I suggest e.g. Antifungal Mechanism of Phenazine-3 1-carboxylic Acid Against Pestalothiopsis kenyana.

2. Keywords duplicate the information in the title this should be changed, e.g. by introducing information about the methods used, e.g. Transcriptomics Analyses, qPCR, SEM, TEM, etc.

3. It is imperative that the abstract provides a transparent account of the research methods employed. It is highly recommended that you revise it accordingly.

4. There are numerous minor  errors in the paper, some of which I have highlighted in the .pdf file. Latin names should be written in italics. Abbreviations should be written with the full name in the first place of the citation.

5. Chapter names should be corrected in several places in both the description of results and the methodology of the paper.

6. For the study Effect of PCA on the Cell Membrane Permeability and Cellular Leakage, statistical analysis should be performed.

7. Subsection 2.7 The report could benefit from a more detailed discussion of the results. The implementation of PCA had an impact on pyruvate metabolism and caused a decline in energy metabolism. It is important to acknowledge the significance of these findings and provide commentary on how they relate to the presented results.

8. Can you provide more details about the research equipment used such as the model, manufacturer, and headquarters? For example, EM (H-7650; Hitachi, Tokyo, Japan)[43]. Additionally, could you please provide a full description of the stereomicroscope and the method used for qRT-PCR Analysis? Also, I am curious to know which method was used to determine the purity and amount of RNA and what concentrations of the matrix were used in the study.

Specific comments:

1. L114 Fig.1-Please state what the letters above the bars on the diagram in part (a) mean. What does the abbreviation CK mean on graph (b) and (c)?

2. I have a question regarding the diagram. Can you explain what the abbreviation CK stands for? Additionally, parts (e) and (f) have red markings that are unclear. Could you please clarify their meaning and provide magnifications for each photo? I am also curious about whether P. kenyana spores were formed in both the control and after PCA application, and what their abundance was. Lastly, did PCA inhibit spore formation?

3. L160- State what is meant by CK.

4. Fig 4- Perform statistical analysis.

5. Fig 6 a and b. Captions on graphs are not legible.

6. L218. GO enrichment analysis- What does GO stand for?

7. L225. KEGG enrichment analysis-What does GO stand for?

8. Fig. 7 a -The captions are not legible this should be corrected.

9. L249. 2.7 qRT-PCR Analysis

In this case, gene expression analysis of pyruvate metabolism in P. kenyana cultures after PCA pledging was determined using the qRT-PCR experimental technique. Therefore, the subsection title should be changed accordingly.

10. Fig. 9. What do the letters a and b above the bars mean?

11. L281-292. Please state what the experimental variants were. Was there a control variant? PCA-287 free solution was used? Please state exactly what was the control in the experiment. Please change this description as it is not very clear at the moment

12. L290- Give a literature citation of the method used in the experiment.

13. L343- Give a citation of the literature used in the method experiment.

14. L360- What was the Transcriptome Analysis involved? Correct the title of the chapter e.g. Transcriptome Analysis of P. kenyana after using PCA

15. L406- Write the names of the genes in italics. What were the efficiencies of the qPCR reactions of the different primer pairs please give them in the table.

16. L408- Statistical Analysis. State the level of significance. Regression analysis was also performed, describe this method in this subsection.

Author Response

Responses to the Reviewer 3

Dear reviewer, we are so appreciated your earnest review and great advice. We have taken care of all the revisions accordingly.

  1. I think you should consider changing the title of the paper because it is long and contains redundant information. I suggest e.g. Antifungal Mechanism of Phenazine-3 1-carboxylic Acid Against Pestalothiopsis kenyana.

Responses:  Thanks for your valuable comments and suggestions. It has been modified in the manuscript according to your comments.

  1. Keywords duplicate the information in the title this should be changed, e.g. by introducing information about the methods used, e.g. Transcriptomics Analyses, qPCR, SEM, TEM, etc.

Responses:  Thanks for your valuable comments and suggestions. It has been modified in the manuscript according to your comments. we have revised the keyword to Bayberry disease, Bioactivity, Mechanism, Pestalotiopsis kenyana, Transcriptomics Analyses, Biogenic fungicide.

  1. It is imperative that the abstract provides a transparent account of the research methods employed. It is highly recommended that you revise it accordingly.

Responses:  Thanks for your valuable comments and suggestions. It has been modified in the manuscript according to your comments.

Pestalotiopsis sp. is an important class of plant pathogenic fungi that can infect a variety of crops. We have proved the pathogenicity of P. kenyana on bayberry leaves and caused bayberry blight. Phenazine-1-carboxylic acid (PCA) has the characteristics of high efficiency, low toxicity, and environmental friendliness, which can prevent fungal diseases on a variety of crops. In this study, the effect of PCA on morphological, physiological, and molecular characteristics of P. kenyana has been investigated, and the potential antifungal mechanism of PCA against P. kenyana was also explored. We applied PCA on P. kenyana in vitro and in vivo to determine its inhibitory effect on PCA. It was found that PCA had a highly efficient against P. kenyana with EC50 around 2.32 μg/mL, and the in vivo effect was 57% at 14 μg/mL. The mechanism of PCA was preliminarily explored by transcriptomics technology. The results showed that after the treatment of PCA, 3613 differential genes were found, focusing on redox processes and various metabolic pathways. In addition, it can also cause mycelial development malformation, damage cell membranes, reduce mitochondrial membrane potential, and increase ROS levels. This result expanded the potential agricultural application of PCA and revealed the possible mechanism against P. kenyana.

  1. There are numerous minor errors in the paper, some of which I have highlighted in the .pdf file. Latin names should be written in italics. Abbreviations should be written with the full name in the first place of the citation.

Responses:  Thanks for your valuable comments and suggestions. It has been modified in the manuscript according to your comments. Besides, we checked all the typos and other errors.

  1. Chapter names should be corrected in several places in both the description of results and the methodology of the paper.

Responses:  Thanks for your valuable comments and suggestions. We have changed the name of the corresponding chapter in the manuscript.

  1. For the study Effect of PCA on the Cell Membrane Permeability and Cellular Leakage, statistical analysis should be performed.

Responses:  Thanks for your valuable comments and suggestions. Statistical analysis was performed using IBM SPSS Statistics 26 for experimental data. Data are mean ±SD and annotated with error bars in the figure 4a,bc, Lines with the “a,b,c” indicate significant differences (p < 0.05) for different concentration.

Figure 4. Effects of PCA on cell membrane permeability and cell leakage of P. kenyana. (a) Cell membrane permeability. (b) The absorbance value of nucleic acids (OD260). (c) The absorbance value of protein (OD280). Data are displayed as the mean ± SD of three replications. Lines with the “a,b,c” indicate significant differences (p < 0.05) for different concentration.

  1. Subsection 2.7 The report could benefit from a more detailed discussion of the results. The implementation of PCA had an impact on pyruvate metabolism and caused a decline in energy metabolism. It is important to acknowledge the significance of these findings and provide commentary on how they relate to the presented results.

Responses:  Thanks for your valuable comments and suggestions. It has been modified in the manuscript according to your comments. The modifications are as follows:

  qRT-PCR assay was performed on four genes related to pyruvate metabolism, including erg10, cut6, lys1, and acu-6, and the result is shown in Figure 9. Among them, only the expression of the acu-6 gene was up-regulated, and the expression of the remaining three genes was down-regulated. This is consistent with transcriptome sequencing results and confirms the reliability of transcriptomics data. This result indicated PCA had an impact on pyruvate metabolism and caused a decline in energy metabolism.

  1. Can you provide more details about the research equipment used such as the model, manufacturer, and headquarters? For example, EM (H-7650; Hitachi, Tokyo, Japan)[43]. Additionally, could you please provide a full description of the stereomicroscope and the method used for qRT-PCR Analysis? Also, I am curious to know which method was used to determine the purity and amount of RNA and what concentrations of the matrix were used in the study.

Responses:  Thanks for your valuable comments and suggestions. It has been modified in the manuscript according to your comments. We have added details of the equipment used in the manuscript. The stereomicroscope details used for the experiment are VHX-6000, Keyence, Osaka, Japan.

The method used for qRT-PCR Analysis: Mycelial plugs (5 mm diameter) from the leading edge of a 5-day-old colony of P. kenyana on potato dextrose agar (PDA) were transferred to potato dextrose broth (PDB) and cultured on a shaking incubator (Shanghai Tiancheng Experimental Instrument Manufacturing Co., Ltd., Shanghai, China) at 25°C, 180 rpm for 3 days, at which PCA was added to a final concentration of 3 µg/ml. Mycelial without PCA was used as the blank control. Then take 0.1g of the hyphae rinsed with PBS and place it in a centrifuge tube. the total RNA of P. kenyana mycelia treated with PCA (3 μg/mL) was extracted by the OMEGA Fungal RNA Kit (Omega Bio-Tek, Guangzhou, China). Then, RNA was reverse transcribed into cDNAs according to the M-MLV First Strand cDNA Synthesis Kit (Omega Bio-Tek, Guangzhou, China). Finally, the perfect start SYBR Green qPCR master mix was used for amplification. The reaction conditions are 95 °C for 10 mins hold, 95 °C for 5 s, 60 °C for 20 s, 72°C for 20 s, and amplification for 40 cycles. Actin was used as an internal control for normalization. The The relative expression level of each gene was calculated by 2−ΔΔCT method.

We use Ultra-micro spectrophotometer (NanoPhotometer N50, Munich, Germany) for measurements of RNA purity. Add 1 μl of RNA dropwise to Ultra-micro spectrophotometer, and use the same amount of DEPC-treated water as a control to read the data.

  1. L114 Fig.1-Please state what the letters above the bars on the diagram in part (a) mean. What does the abbreviation CK mean on graph (b) and (c)?

Responses:  Thanks for your valuable comments and suggestions. Different letter labels indicate significant differences between different treatment groups. The CK on graph (b) and (c) means control group without PCA.

  1. I have a question regarding the diagram. Can you explain what the abbreviation CK stands for? Additionally, parts (e) and (f) have red markings that are unclear. Could you please clarify their meaning and provide magnifications for each photo? I am also curious about whether P. kenyana spores were formed in both the control and after PCA application, and what their abundance was. Lastly, did PCA inhibit spore formation?

Responses:  Thanks for your valuable comments and suggestions. The abbreviation CK means control group without PCA. The red markings in parts (e) and (f) means mitochondria(Mt) and lipid droplet(Ld). The magnification of TEM is 1.0 μm. The magnification of SEM is 10.0 μm. The magnification of stereomicroscope is 1500 times. Unfortunately, we did not conduct experiments on the effect of PCA on Pestalotiopsis kenyana spore formation.

  1. L160- State what is meant by CK.

Responses:  Thanks for your valuable comments and suggestions. The CK means treatment group without PCA.

  1. Fig 4- Perform statistical analysis.

Responses:  Thanks for your valuable comments and suggestions. Statistical analysis was performed using IBM SPSS Statistics 26 for experimental data,data results are mean ±SD and annotated with error bars in the figure.

  1. Fig 6 a and b. Captions on graphs are not legible.

Responses:  Thanks for your valuable comments and suggestions. It has been modified in the manuscript according to your comments. We have increased the resolution.

  1. L218. GO enrichment analysis- What does GO stand for?

Responses:  Thanks for your valuable comments and suggestions. The full name of GO (Gene Ontology) was given. Three types of descriptions are provided for systematic definition of the function of gene products. The structure of GO includes three aspects: Molecular Function, Biological Process, and Cellar Component in the cell.

  1. L225. KEGG enrichment analysis-What does GO stand for?

Responses:  Thanks for your valuable comments and suggestions. The full name for KEGG  were given (Kyoto Encyclopedia of Genes and Genomes) as it appeared first time. It is public database for genome deciphering. In organisms, different genes coordinate with each other to perform their biological functions, and Pathway-based analysis helps to further understand the biological functions of genes. KEGG is the main public database on the pathway.

  1. Fig. 7 a -The60 are not legible this should be corrected.

Responses:  Thanks. We have increased the resolution in the revised MS by listing a and b seprately

  1. L249. 2.7 qRT-PCR Analysis

In this case, gene expression analysis of pyruvate metabolism in P. kenyana cultures after PCA pledging was determined using the qRT-PCR experimental technique. Therefore, the subsection title should be changed accordingly.

Responses:  Thanks for your valuable comments and suggestions. It has been modified to Gene Expression Analysis in the manuscript according to your comments.

  1. Fig. 9. What do the letters a and b above the bars mean?

Responses:  Thanks for your valuable comments and suggestions. A description was given as "Data are displayed as the mean ± SD of three replications. Different letters indicate significant differences."

  1. L281-292. Please state what the experimental variants were. Was there a control variant? PCA-287 free solution was used? Please state exactly what was the control in the experiment. Please change this description as it is not very clear at the moment

Responses:  Thanks for your valuable comments and suggestions. It has been modified in the manuscript according to your comments. CK means the data from the control group that normal P. kenyana. (with no PCA treatment).

In the in vivo activity test, we used 1% PCA Suspension Concentrate (purchased from Good Harvest Company, Shanghai, China), using sterile water as a control, and the experimental subject was a biennial Dongkui bayberry and kept in an artificial climate chamber(25℃, humidity 80%).

  1. L290- Give a literature citation of the method used in the experiment.

Responses:  Thanks for your valuable comments and suggestions. It has been modified in the manuscript line 291 according to your comments.

  1. L343- Give a citation of the literature used in the method experiment.

Responses:  Thanks for your valuable comments and suggestions. The citation to the method reference ([45]) is located in line 337.

  1. L360- What was the Transcriptome Analysis involved? Correct the title of the chapter e.g. Transcriptome Analysis of P. kenyana after using PCA

Responses:  Thanks for your valuable comments and suggestions. We have modified the title to Transcriptome Analysis of PCA Against P. kenyana as your comments in the manuscript.

  1. 23. L406- Write the names of the genes in italics. What were the efficiencies of the qPCR reactions of the different primer pairs please give them in the table.

Responses:  Thanks for your valuable comments and suggestions. The gene name has been changed to italics in the manuscript according to your comments. The qPCR reactions of the different primer pairs were given in the table shown below.

Gene_ID

log2FC

Primer Sequences (5’-3’)

Primer’s efficiency

acu-6

2.03

F- TGTATACGAAGTCGGTTCAGCC

R- TAACCGTCAACGACGTAAATTC

96

erg10

-1.73

F- TCCAACACCCCGCACTACCTGC

R- GCTGACCTCGACGGGGACGATC

95

lys1

-1.31

F- GTGAACAGTTCGCCAATGCCTA

R- AGAGTGCTCGCGGAGGTGCTG

97

cut6

-1.46

F- AGCAGCCTCCCCAGGGTGAAT

R- TCGGTACACTTGTTGAAGAAGTGG

97

actin

F- CTGGCACCGTCGTCGATGAAG

R- AAGGTCCGCTCTCGTCGTACTC

99

  1. L408- Statistical Analysis. State the level of significance. Regression analysis was also performed, describe this method in this subsection.

Responses:  Thanks for your valuable comments and suggestions. It has been modified in the manuscript according to your comments. The pagrapha was modified as "All data analyses were performed using IBM SPSS Statistics 26 Single factor AVONA test and Duncan analysis were used to analyze and compare the data obtained. Data are averages ±SD. It is represented by an error bar in the figure."

Round 2

Reviewer 2 Report

Dear Authors,

I am satisfied with the corrections made to the manuscript and the responses to my questions and concerns. After re-reading the manuscript, I have minor editorial comments:

1.    Please transfer caption of the figure 5 from page 7 to 6. Manuscript will look better, captions should always accompany figures on the same page.

2.    For Figure 6, please standardize the size of graphs a and b. The axes should be roughly the same size for better visual perception.

Considering the above arguments, I recommend this manuscript for publication in International Journal of Molecular Sciences in present form with this 2 improvements.

Reviewer 3 Report

All amendments have been made. I recommend that the manuscript be accepted for publication. I congratulate the authors on a job well done.